Integrative systematic revision of the Montseny brook newt (Calotriton arnoldi), with the description of a new subspecies

Talavera Adrián 1 adrian.talavera@csic.es
Valbuena-Ureña Emilio 2
http://orcid.org/0000-0001-9832-1782 Burriel-Carranza Bernat 1 3
Mochales-Riaño Gabriel 1
Adams Dean C. 4
Amat Fèlix 5
Carbonell Francesc 6
Carranza Salvador 1
1 Institute of Evolutionary Biology (CSIC-Universitat Pompeu Fabra) , Barcelona, Catalonia , Spain
2 Servei de Gestió de Parcs Naturals. Diputació de Barcelona , Barcelona , Spain
3 Museu de Ciències Naturals de Barcelona , Barcelona, Catalonia , Spain
4 Department of Ecology, Evolution, and Organismal Biology, Iowa State University , Ames, Iowa , United States
5 Àrea d’Herpetologia, Museu de Granollers—Ciències Naturals , Granollers, Catalonia , Spain
6 Centre de Fauna Salvatge de Torreferrussa , Santa Perpètua de Mogoda, Catalonia , Spain
Brannelly Laura
Electronic publication date: 2024 Jun 13
Publication date: 2024
Volume: 12
Electronic Location ID: e17550
Received 2024 Jan 30; Accepted 2024 May 20
Copyright: © 2024 Talavera et al.
Copyright year: 2024
Copyright holder: Talavera et al.
License: This is an open access article distributed under the terms of the Creative Commons Attribution License, which permits unrestricted use, distribution, reproduction and adaptation in any medium and for any purpose provided that it is properly attributed. For attribution, the original author(s), title, publication source (PeerJ) and either DOI or URL of the article must be cited.
License URL: https://creativecommons.org/licenses/by/4.0/

Keywords: Amphibians, Montseny brook newt, Calotriton arnoldi laietanus ssp. nov., Morphology, Conservation, Subspecies

Funding: “la Caixa” doctoral fellowship programme LCF/BQ/DR20/11790007 Ministerio de Ciencia, Innovación y Universidades, Spain FPU18/04742 and PRE2019-088729 Diputació de Barcelona—Àrea d’Espais Naturals Generalitat de Catalunya 2021 SGR 00751 LIFE project LIFE15 NATO/SE/000757 Adrián Talavera is supported by “la Caixa” doctoral fellowship programme (LCF/BQ/DR20/11790007). Bernat Burriel-Carranza was funded by FPU grant from Ministerio de Ciencia, Innovación y Universidades, Spain (FPU18/04742). Gabriel Mochales-Riaño was funded by an FPI grant from the Ministerio de Ciencia, Innovación y Universidades, Spain (PRE2019-088729). The study was developed under the funding of the Diputació de Barcelona—Àrea d’Espais Naturals, and the grant 2021 SGR 00751 from the Departament de Recerca i Universitats from Generalitat de Catalunya. The LIFE project (LIFE15 NATO/SE/000757) financially supported some authors of this work. The funders had no role in study design, data collection and analysis, decision to publish, or preparation of the manuscript.

==============================
The Montseny brook newt (Calotriton arnoldi), a glacial relict endemic to a small, isolated massif in northeast Spain, is considered the only Critically Endangered urodele in Europe. Its restricted range is divided by a deep valley that acts as an impassable barrier to dispersal, separating two isolated metapopulations (Western and Eastern) that correspond to independent lineages with different evolutionary trajectories, based on genetic and genomic data. Here, we address the ecological differentiation between lineages and discuss its potential effect on the phenotypic distinctness of each lineage. Based on multiple lines of evidence, we formally describe the Western Montseny brook newt as a new subspecies: Calotriton arnoldi laietanus ssp. nov. Finally, our study underscores the importance of considering taxonomic progress in the conservation policies of endangered species, ensuring appropriate management and protection of the newly described taxa.

Introduction

Climatic oscillations throughout the Pleistocene have shaped biodiversity clines in the Palearctic and, more specifically, in Europe, with northern populations often showing low morphological and genetic variability in contrast to well-differentiated southern populations (Hewitt, 2000). This biogeographical pattern, resulting from successive cycles of extinction and recolonization processes in the north versus population persistence in southern refugia, has parallels at smaller spatial scales, as exemplified by the brook newts of the genus Calotriton Gray, 1858. This group of caudate amphibians is endemic to the Pyrenees and neighboring lesser mountain ranges, and comprises two species: the relatively widespread Pyrenean brook newt, C. asper, and the geographically restricted Montseny brook newt, C. arnoldi, secluded in a few isolated creeks in the Montseny Massif, located to the southeast of the Pyrenees (Catalonia, NE Spain; Fig. 1) (Carranza & Amat, 2005). Endemic taxa with small ranges are more prone to extinction due to stochastic phenomena than widespread species (Allendorf, Luikart & Aitken, 2012), but in addition, the Montseny brook newt faces a number of major threats, including global warming, the outbreak of lethal emergent diseases, human-induced habitat fragmentation and rampant groundwater depletion in their primary aquatic habitats (Carranza & Amat, 2005; Martel et al., 2020; IUCN SSC Amphibian Specialist Group, 2022; Talavera et al., 2024). Consequently, this microendemism has been listed as the only Critically Endangered urodele in Europe (IUCN SSC Amphibian Specialist Group, 2022). Their natural populations, which have been estimated to comprise around 1,000–1,500 individuals (IUCN SSC Amphibian Specialist Group, 2022), are currently being reinforced with an ex-situ breeding program, that introduces the species in suitable but unoccupied habitats in an attempt to increase the area inhabited by the species (Guinart et al., 2022), which is estimated to be barely 10 km2 (IUCN SSC Amphibian Specialist Group, 2022).

Figure 1 Distribution of the brook newts of the genus Calotriton in the Pyrenees and neighboring mountain ranges, in southwestern Europe.

Whereas the Pyrenean brook newt Calotriton asper (in grey) is relatively widespread, the Montseny brook newt (Calotriton arnoldi) is a microendemism exclusive to the Montseny Massif, located southeast to the Pyrenees in Catalonia, Spain. This massif is split into two sectors from West to East, dividing two genetic and genomic lineages of the species, herein formally described as the Western Montseny brook newt C. arnoldi laietanus ssp. nov. (blue) and the Eastern Montseny brook newt C. arnoldi arnoldi (orange). Ranges modified from Talavera et al. (2024).

Despite its small and restricted natural distribution, deep genetic and genomic differences have been found in populations across the Tordera River Valley, which divides the Montseny Massif into two sectors (West and East, Carranza & Amat, 2005; Valbuena-Ureña et al., 2017a; Talavera et al., 2024). Consequently, two major lineages are isolated on each side of the valley (Fig. 1): the Eastern Montseny lineage comprises a somewhat fragmented metapopulation distributed in three brooks, with signs of genomic differentiation among them, whereas the Western submassif is inhabited by a larger metapopulation occupying five brooks and displaying high connectivity (Talavera et al., 2024). There is no evidence for current connectivity among lineages, but they seem to have been weakly connected in the past (Talavera et al., 2024). Although the two lineages are just a few kilometers apart, migration across the Tordera River Valley is hampered by biological constraints, including the extremely low dispersal capacity of the species (mean displacements of 7 m in a 2-year period, Guinart et al., 2022), and the lack of functional lungs, which restricts dispersal to water-way routes. The minimum aquatic distance between lineages exceeds the 30 km, crossing unsuitable areas in terms of water temperature, pollution, and presence of predatory fish. However, genomic differences between lineages cannot be exclusively attributed to isolation by distance (Talavera et al., 2024).

Levels of genomic differentiation are sufficient to consider each lineage as a valid candidate species following Hausdorf & Hennig (2020). The ongoing conservation program has considered them as two different evolutionary significant units from the onset (Valbuena-Ureña, Amat & Carranza, 2013), in the absence of formal subspecific designations. In fact, the Montseny brook newt lineages meet some of the proposed lines of evidence to support any species hypothesis under the unified species concept (De Queiroz, 1998, 2007), such as genetic and genomic monophyly (Carranza & Amat, 2005; Valbuena-Ureña, Amat & Carranza, 2013; Talavera et al., 2024). However, other lines of evidence remain unexplored. Here, we assessed ecological differentiation in the disjoint Montseny brook newt ranges at opposing sides of the Tordera River Valley, and its possible relationship with previously described differences in coloration between lineages (Valbuena-Ureña, Amat & Carranza, 2013). In addition, we re-analyzed published morphological data to delve into their phenotypical differentiation. Finally, through the integration of genomic, morphological, and ecological evidence of the distinctiveness of both lineages, we conclude that the Montseny brook newt lineages deserve taxonomic recognition at the subspecific rank, and formally describe the Western Montseny brook newt as a new subspecies.

Materials and Methods

Ecological analyses

We used 596 exact points (471 from Western Montseny and 125 from Eastern Montseny) with presence data for the Montseny brook newt (C. arnoldi) to assess ecological differences among lineages by extracting and statistically analyzing associated environmental data from different sources with QGIS v3.22.1 (QGISorg, 2022). Fieldwork was authorized in Catalonia by the Departament ďAcció Climàtica, Alimentació i Agenda Rural of the Catalan Government with the permission numbers SF/298 and SF/469. Nineteen bioclimatic variables were retrieved from Worldclim v2 (Fick & Hijmans, 2017) with a resolution of 30 s (~1 km2) and merged into two summary variables using a principal component analysis (PCA), one related to temperature (PC1 of BIO1-11; Table 1) and another to precipitation (PC1 of BIO12-19; Table 2). Altitude data was retrieved from Copernicus EU-DEM v.1.1 (https://land.copernicus.eu/imagery-in-situ/eu-dem/eu-dem-v1.1), with a resolution of ~25 m and data on potential natural vegetation (“Phytosociological alliances”) was obtained from the Catalonian Vegetation Map MVC50 (Carrillo et al., 2018), with a resolution of ~50 m. Exact locations are not disclosed for conservation reasons. To statistically test for differences in temperature PC1, precipitation PC1, and altitude between the ranges of both Montseny lineages, we carried out, for each variable, 10,000 one-way ANOVA tests subsampling 250 records from each submassif (West and East) and allowing replacement from the above-mentioned dataset. The distribution of these F values was compared to a null distribution of 10,000 lineage-blind ANOVA F values, performed subsampling 500 random records of the species (not differentiating by submassif), using Student’s t-tests.

Table 1 Factor loadings of the temperature-related variables that summarize differences between Montseny subspecies ranges.

Temperature PC1	Factor loadings	
Annual mean temperature	0.158	
Annual diurnal range	0.020	
Isothermality	0.072	
Temperature seasonality	0.749	
Max temperature of warmest month	0.132	
Min temperature of coldest month	0.125	
Temperature anual range	0.007	
Mean temperature of wettest quarter	0.553	
Mean temperature of driest quarter	0.164	
Mean temperature of warmest quarter	0.140	
Mean temperature of coldest quarter	0.152	
Note:

Main loadings in bold.

Table 2 Factor loadings of the precipitation-related variables that summarize differences between Montseny subspecies ranges.

Precipitation PC1	Factor loadings	
Annual precipitation	0.839	
Precipitation of wettest month	0.070	
Precipitation of driest month	0.031	
Precipitation seasonality	−0.017	
Precipitation of wettest quarter	0.267	
Precipitation of driest quarter	0.311	
Precipitation of warmest quarter	0.006	
Precipitation of coldest quarter	0.349	
Note:

Main loadings in bold.

An analogous procedure was carried out to test for differences in botanical communities between submassifs. For this purpose, phytosociological alliances were used as proxies of simplified forest types: Fagion sylvaticae as beech forest, Quercion ilicis as evergreen oaks, Alnion incanae and Populion albae as riparian communities, and Carpinion as deciduous oaks. The Antirrhinion asarinae, Galeopsion pyrenaicae, Galeopsion segetum, Phagnalo-Cheilanthion and Pimpinello-Gouffeion alliances were instead considered simply as rock outcrops. We retained χ² values from 10,000 tests in which we had subsampled 250 points for each submassif allowing replacement. The distribution of these χ² values was compared to a null distribution obtained from the same number of lineage-blind χ² tests by a Student’s t-test.

Morphological analyses

We re-analyzed a morphology dataset from 160 wild adult animals (86 males and 74 females), including both lineages (62 Western Montseny and 98 Eastern Montseny; from populations B1, B2, B3 and A1), from Valbuena-Ureña, Amat & Carranza (2013). Eight linear morphometric measurements, obtained using digital calipers, were analyzed, namely snout-vent length (SVL), head length (HL), head width (HW), forelimb and hindlimb lengths (FLL & HLL), limb interval (LI), tail length (TL), and tail height (TH). Normality of variables was tested with Q-Q plots and mean and standard deviation were calculated for each sex*lineage category. For subsequent analyses, all measurements were log-transformed and, contrary to Valbuena-Ureña, Amat & Carranza (2013), we used SVL as a proxy of individual size, to analyze the shape independently from size. For this, we extracted the residuals of regressions of the seven remaining variables against size and performed a PCA. The influence of sex, lineage and their interaction on each linear measurement, and of PC1 and PC2 of shape was tested using a linear model in the R package RRPP (Collyer & Adams, 2018), and ANOVA statistics were evaluated by generating empirical sampling distributions based on residual randomization using 1,000 permutations. Finally, the PC scores of shape were plotted in violin plots for the four categories (two lineages, both sexes) with the R package ggplot2 v3.3.5 (Wickham, 2016).

Taxonomic description

A set of 12 morphometric measurements were taken with digital calipers (rounding to the nearest 0.1 mm) to describe the holotype of the newly described taxon. The electronic version of this article in Portable Document Format (PDF) will represent a published work according to the International Commission on Zoological Nomenclature (ICZN), and hence the new names contained in the electronic version are effectively published under that Code from the electronic edition alone. This published work and the nomenclatural act it contains have been registered in ZooBank, the online registration system for the ICZN. The ZooBank LSIDs (Life Science Identifiers) can be resolved and the associated information viewed through any standard web browser by appending the LSID to the prefix http://zoobank.org/. The LSID for this publication is: urn:lsid:zoobank.org:pub:AE201B51-F9B7-4B7F-A70E-D5FCC05EB335. The online version of this work is archived and available from the following digital repositories: PeerJ, PubMed Central SCIE and CLOCKSS.

Results

Ecological and morphological differences

The Western and Eastern Calotriton arnoldi lineages inhabit ranges that significantly differ in altitude (t19998 = 602.87, p << 0.001), temperature (PC1: 99.59% variance; t19998 = 559.66, p << 0.001), and precipitation regimes (PC1: 95.84% variance; t19998 = 638.43, p << 0.001). The Eastern Montseny brook newt lineage inhabits higher areas (Mean East: 1,085 m above sea level (m.a.s.l.); mean West: 887 m.a.s.l), with a colder wet season (Mean East: 11.6 °C; mean West: 13.7 °C), slightly higher temperature seasonality (SD East: 5.74 °C; SD West: 5.71 °C), and with a higher mean annual rainfall (East: 927 mm; West: 874 mm). These differences are associated with the presence of dissimilar forest assemblages (t19998 = 1,817.1; p << 0.001, frequencies per community and lineage range in Table 3). The Western Montseny brook newt lineage occupies streams covered mostly by holm oak (Quercus ilex) forests and, less frequently, by sessile oaks (Q. petraea) or beeches (Fagus sylvatica). On the other hand, the Eastern Montseny brook newts inhabit almost exclusively streams in either beech or riparian forests (i.e., Populus spp., Alnus glutinosa). Dissimilar habitats are associated with contrasting color patterns, which are fixed in each submassif (Valbuena-Ureña, Amat & Carranza, 2013; Figs. 2A, 3, 4A).

Table 3 Percentages of C. arnoldi habitat covered by each type of botanical community by subspecies.

Subspecies	Beech forests	Deciduous oaks	Evergreen oaks	Riparian communities	Rock outcrops	
C. arnoldi laietanus ssp. nov.	17.0	22.3	56.3	0.2	4.2	
C. arnoldi arnoldi	47.2	0.0	2.4	50.4	0.0	
Note:

In the East, beech and riparian forests prevail, whereas in the West, oaks (mainly holm oaks) are dominant.

Figure 2 Phenotypical differences between Montseny brook newt lineages.

(A) Pictures of adult female representatives of each subspecies. The more robust Western Montseny brook newt Calotriton arnoldi laietanus ssp. nov. always exhibits silvery-golden stippling on its flanks (see Fig. 4A). The slenderer Eastern Montseny brook newt Calotriton arnoldi arnoldi usually shows, in contrast, yellow blotches on the sides of the tail and body. (B and C) Morphology PCA results depicted as violin plots per sex and lineage for the first (B) and second (C) principal components. PC1 (36.28% of the shape variance) represents sexual dimorphism in the species regardless of the lineages, whereas PC2 (16.49%) is significantly associated with differences between the subspecies, i.e., length of limbs, width of the head and length and height of the tail. See Tables 5 and 6. Photo credits: Adrián Talavera.

Figure 3 Ecological differences between the Western and Eastern Montseny brook newt ranges regarding botanical assemblages.

Pictures of representatives of each subspecies on top: C. arnoldi laietanus ssp. nov. (left) and C. arnoldi arnoldi (right); and percentages (bottom) of each type of forest covering the inhabited brooks: evergreen oaks (darkest green), deciduous oaks (dark green), beech forest (green), riparian forest (light green) or rock outcrops (grey). See Table 3. Western Montseny brook newts inhabit brooks in evergreen forests with less leaf litter, whereas the Eastern conspecifics inhabit brooks in deciduous forests with a high amount of leaves, in which their chocolate skin hues and yellowish blotches probably confer a better camouflage. Photo credits: Adrián Talavera.

Figure 4 Differential diagnosis of the newly described subspecies.

(A) The Western Montseny brook newt C. a. laietanus ssp. nov. (adult male on the left) always exhibit silvery-gold stippling on the flanks, and, often, white snout margins, especially older males. Both characteristics are never shown by the Eastern Montseny brook newt C. arnoldi arnoldi (adult male on the right, exhibiting a couple of melanophoromas on the body (Martínez-Silvestre et al., 2011)). (B) Habitat of the Western Montseny brook newt: small brooks in humid gullies, covered mainly by holm oaks (Quercus ilex) and facing northwards. In contrast, Eastern Montseny brook newts typically inhabit brooks in beech (Fagus sylvatica) or riparian forests, at higher altitudes and facing south. Photo credits: Adrián Talavera.

Mean and standard deviation of SVL and the seven shape linear measurements for each sex*lineage category are shown in Table 4, and two-factor ANOVA results for the same variables and categories in Table 5. Factor loadings of the two first components in the morphology PCA are shown in Table 6. The first component (35.98% of variance) explained sexual dimorphism in shape for both Montseny lineages, whereas the second one (16.77%) explained differences between lineages (Figs. 2B, 2C). Eastern Montseny brook newts have proportionally longer limbs and higher and longer tails, whereas the Western Montseny brook newts have wider heads (Tables 4–6). Males from both lineages have relatively longer and wider heads, longer limbs, and shorter tails than females (Tables 4–6). Size does not significantly differ either between sexes (p = 0.094) or lineages (p = 0.812), although their interaction is marginally significant (p = 0.042): Western males are slightly smaller than Western females, while Eastern Montseny brook newts do not differ between sexes (Table 5).

Table 4 Mean ± SD of the linear measurements in millimeters used in morphological analyses per each sex*lineage category.

	Western montseny
Calotriton arnoldi laietanus spp. nov.	Eastern montseny
Calotriton arnoldi arnoldi	
	Males (n = 38)	Females (n = 24)	Males (n = 48)	Females (n = 50)	
	Mean ± SD	Mean ± SD	Mean ± SD	Mean ± SD	
Snout-vent length (SVL)	59.1 ± 2.25	61.0 ± 3.06	59.9 ± 2.25	60.1 ± 3.16	
Head length (HL)	13.0 ± 0.917	12.1 ± 0.85	13.5 ± 0.99	12.1 ± 0.93	
Head width (HW)	10.8 ± 0.79	9.73 ± 0.61	10.3 ± 0.69	9.38 ± 0.67	
Forelimb length (FLL)	14.7 ± 0.98	13.9 ± 1.11	15.6 ± 0.75	14.1 ± 0.63	
Hindlimb length (HLL)	17.1 ± 1.08	16.0 ± 0.85	17.9 ± 0.85	16.2 ± 0.73	
Limb interval (LI)	31.0 ± 2.27	31.3 ± 3.33	31.1 ± 2.89	30.6 ± 2.43	
Tail length (TL)	40.3 ± 1.84	42.2 ± 3.04	43.2 ± 2.04	45.1 ± 2.74	
Tail height (TH)	5.53 ± 1.63	6.50 ± 1.72	6.51 ± 1.64	6.5 ± 1.84	

Table 5 ANOVA statistics for size (SVL) and other seven linear morphometric log-transformed measurements corrected by size, with sex and lineage as factors.

Dep. variable	Ind. variable	SS (type II)	R2	F1	Z	P-value	
Snout-vent length (SVL)	Sex	0.061	0.018	2.99	1.349	0.094	
	Lineage	0.000	0.000	0.052	−0.951	0.812	
	Sex:Lineage	0.008	0.024	3.936	1.628	0.042	
Head length (HL)	Sex	0.419	0.364	90.716	5.478	<0.001	
	Lineage	0.017	0.015	3.667	1.488	0.068	
	Sex:Lineage	0.009	0.008	1.978	1.013	0.165	
Head width (HW)	Sex	0.427	0.367	105.490	5.609	<0.001	
	Lineage	0.058	0.050	14.268	2.727	<0.001	
	Sex:Lineage	0.631	0.002	0.558	0.099	0.480	
Forelimb length (FLL)	Sex	0.308	0.401	114.573	6.035	<0.001	
	Lineage	0.062	0.081	23.039	3.537	<0.001	
	Sex:Lineage	0.006	0.008	2.291	1.146	0.130	
Hindlimb length (HLL)	Sex	0.318	0.447	132.223	5.660	<0.001	
	Lineage	0.030	0.043	12.608	2.697	<0.001	
	Sex:Lineage	0.007	0.010	2.830	1.314	0.100	
Limb interval (LI)	Sex	0.011	0.012	1.831	0.962	0.176	
	Lineage	0.003	0.004	0.565	0.137	0.472	
	Sex:Lineage	0.000	0.000	0.000	−2.415	1.000	
Tail length (TL)	Sex	0.048	0.079	20.644	3.411	<0.001	
	Lineage	0.172	0.280	73.342	5.314	<0.001	
	Sex:Lineage	0.003	0.004	1.173	0.619	0.288	
Tail height (TH)	Sex	0.094	0.007	1.102	0.657	0.266	
	Lineage	0.328	0.023	3.833	1.566	0.049	
	Sex:Lineage	0.255	0.018	2.976	1.315	0.095	
PC1	Sex	265.093	0.662	312.896	7.286	<0.001	
	Lineage	1.497	0.004	1.767	1.003	0.169	
	Sex:Lineage	2.669	0.007	3.151	1.402	0.078	
PC2	Sex	0.278	0.002	0.393	−0.062	0.540	
	Lineage	73.369	0.393	103.530	6.004	<0.001	
	Sex:Lineage	0.269	0.001	0.380	−0.114	0.564	
Note:

Significant factors or interactions shown in bold.

Table 6 Factor loadings of the two main PCs of morphology principal component analysis.

Measurements	PC1	PC2	
Head length	0.481	0.052	
Head width	0.444	0.380	
Forelimb length	0.504	−0.380	
Hindlimb length	0.524	−0.313	
Limb interval	0.029	0.038	
Tail length	−0.204	−0.772	
Tail height	−0.028	−0.120	
Note:

PC1 summarizes sexual dimorphism and PC2 explains the morphological differences between Montseny lineages. Factor loadings of significantly different measurements for sex in PC1 or lineage in PC2 regarding ANOVA tests shown in bold.

Systematic revision of the Western Montseny brook newt lineage

The two Montseny brook newt lineages exhibit genetic and genomic divergence associated with the strong barrier effect imposed by the Tordera River (Carranza & Amat, 2005; Valbuena-Ureña, Amat & Carranza, 2013; Talavera et al., 2024), coupled with subtle morphological differentiation (Valbuena-Ureña, Amat & Carranza, 2013; this study), and fixed color differences linked to ecological differences in climatic and vegetation-related variables. Congruence among these lines of evidence, lead us to recognize the Western submassif lineage as a different subspecies within Calotriton arnoldi, which we formally describe below.

Systematics

Species Calotriton arnoldi Carranza & Amat, 2005

Calotriton arnoldi laietanus ssp. nov. Talavera, Amat, Valbuena-Ureña, Carbonell & Carranza

Western Montseny brook newt. Figs. 2–4, Tables 3–5.

Calotriton arnoldi (partim); Carranza & Amat (2005), Valbuena-Ureña, Amat & Carranza (2013), Speybroeck et al. (2016), Valbuena-Ureña et al. (2017a), Salvador, Pleguezuelos & Reques (2021) Guinart et al. (2022), Talavera et al. (2024).

ZooBank registration

This nomenclatural act is registered in ZooBank. LSID: urn:lsid: zoobank.org:act:5C8988C2-68EE-4AC6-8F67-8602FAA84CD2.

Etymology

No taxonomic name was available for the Western Montseny brook newt lineage, so we coined a new name based on the demonym after the Laietani, an ancient pre-Roman Iberian people for whom the Tordera River represented, as well, their easternmost frontier. As a common name we propose the Western Montseny brook newt, in contrast to the nominal subspecies, henceforth referred as the Eastern Montseny brook newt. We emphasize the difference with the Western brook newts, as present-day Calotriton spp. were commonly known when included within the genus Euproctus Genè, 1839 (see Carranza & Amat (2005)).

Type material

Holotype of C. arnoldi laietanus ssp. nov. deposited at Museu de Ciències Naturals de Barcelona (MCNB), with accession number: MZB 2024-2573. Adult female preserved in a 4% formaldehyde solution. Born in the wild around 2002 (estimated with skeletochronology), captured to become a founder of the ex-situ breeding program for the species from population B1 in the Western Montseny submassif. Deceased in captivity at Centre de Fauna Salvatge de Torreferrussa (Santa Perpètua de Mogoda, Spain) on November 20th, 2023, collected by Emilio Valbuena-Ureña. Twenty-four microsatellite loci from the holotype have been analyzed in Valbuena-Ureña et al. (2017b), coded as sample 39.

Paratypes of C. arnoldi laietanus ssp. nov.: Specimen MZB 2024-2575. Adult male preserved in a 4% formaldehyde solution, deposited at MCNB. Born in the wild around 2003 (estimated with skeletochronology), capture to become a founder of the ex-situ breeding program for the species from population B1 in the Western Montseny submassif. Deceased in captivity at Centre de Fauna Salvatge de Torreferrussa (Santa Perpètua de Mogoda, Spain) on November 20th, 2023, collected by Emilio Valbuena-Ureña. Twenty-four microsatellite loci from this specimen have been analyzed in Valbuena-Ureña et al. (2017b), coded as sample 34.

Specimen MZB 2024-2574. Adult female preserved in 4% formaldehyde solution, deposited at MNCB. Born in captivity in 2014, descendant of wild founder individuals (coded TOC21 & TOC24) from populations B1 and B2, at Centre de Fauna Salvatge de Torreferrussa. Collected by Emilio Valbuena-Ureña on January 9th, 2024, after dying from natural causes.

Specimen MCNG 13809. Adult female preserved in ethanol, deposited at the Museu de Granollers—Ciències Naturals (Barcelona, Spain). Wild individual found dead after a storm at population B4 from the Western Montseny submassif, collected by Fèlix Amat on November 26th, 2019.

Diagnosis

A new subspecies of Montseny brook newt endemic to the western bank of the Tordera River Valley in the Montseny Natural Park, Catalonia, northeast Spain, characterized by the combination of the following characters: 1.–medium-sized newt with slightly larger females (maximum size (SVL + TL) in mm: female = 124.1 (72.9 + 51.2), male = 116.5 (68 + 48.5); 2.–silvery-gold stippling on the flanks, always dorsally uniform in brown or dark-brown color; 3.–a thin brownish-orange dorsal stripe from the base to the tip of the tail, sometimes extending into the body; 4.–snout of all males and occasionally some females with white margins.

Differential diagnosis

Calotriton arnoldi laietanus ssp. nov. differs from its phylogenetically closely-related subspecies C. a. arnoldi by being slightly larger (maximum size (SVL + TL) in mm: female = 124.1 (72.9 + 51.2), male = 116.5 (68 + 48.5) versus maximum size Eastern female: 115.4 (66 + 49.5), male: 110.5 (63 + 47.5)); by being more robust, with proportionally shorter and less dorso-ventrally expanded tails, shorter limbs, and wider heads; by having a uniform brown or dark-brown color with silvery-gold stippling on the flanks versus presence of light greenish or yellowish blotches on the sides of the tail and body over a brown or dark-brown color in Eastern populations (78.8% of individuals, Valbuena-Ureña, Amat & Carranza (2013)) (see Figs. 2A, 3, 4A). Those blotches are more obvious in juveniles and can be exceptionally present in juvenile specimens of C. a. laietanus ssp. nov. but never in adults; and by having white snout margins versus snout with the same body color (see Fig. 4A).

Biogeographical and ecological remarks

All populations from the western bank of the Tordera River (i.e., B1–B5) belong to the newly described taxon. Ecologically, Western Montseny brooks are present at lower altitudes, in streams covered typically by holm oaks, facing north and in warmer and less pluvious localities than their conspecifics’ habitats (Fig. 4B). Calotriton a. arnoldi typically inhabits brooks at higher altitudes, in south-facing beech and other deciduous broad-leaved forests, comprising the natural populations on the eastern bank of Tordera River Valley (populations A1–3).

Genetic and genomic remarks

According to Valbuena-Ureña, Amat & Carranza (2013), Western Montseny brook newts share unique haplotypes for two different markers: the mitochondrial gene cyt b and the nuclear gene RAG-1 (GenBank accession numbers: KC665954 and KC665966, respectively). The cytochrome b Western Montseny haplotype differs by 2–4 mutations from the Eastern haplotypes, whereas only 1–2 mutations separate the RAG-1 haplotypes of the two lineages. Regarding genomics (ddRADseq data), fixation index between lineages reaches up to 80.1% (i.e., the percentage of retrieved diversity within the species that is fixed in either C. a. laietanus ssp. nov. or in C. a. arnoldi), a higher amount of divergence than expected from water-way geographical distances if divergence was merely the result of isolation by distance (Talavera et al., 2024).

Conservation remarks

The identification of two subspecies within the Critically Endangered Calotriton arnoldi (IUCN SSC Amphibian Specialist Group, 2022) should encourage conservation efforts in order to prevent the loss of such unique lineages. Regarding the newly described taxon, Calotriton arnoldi laietanus ssp. nov., its effective population size appears to be larger than in its conspecific C. a. arnoldi, and comprises more populations with slightly higher genomic diversity (Talavera et al., 2024). Unfortunately, the conservation status of the nominotypical subspecies C. a. arnoldi is more concerning due to several converging stressors, such as stronger isolation of demes, human-induced fragmentation, smaller population sizes and dependence on a more vulnerable habitat to persist in the face of rising temperatures (Talavera et al., 2024).

Holotype description

An adult female with head strongly dorsoventrally depressed, broadest at level of the posterior margin of parotid area. Blunt snout with canthus rostralis and prominent swellings on the posterior sides of the head. Upper lip margin white. Slightly dorsoventrally depressed body; oval cross-section; tubercles tipped with hard blunt spines widely distributed on head, tail, and body, especially on flanks. Cylindrical cloaca, relatively narrow. Digits on fore- and hindlimbs 4:5, not elongated, unwebbed, with dark tips. Short tail (~76% of SVL), laterally compressed, and slightly dorsally expanded (~16% of tail length at maximum tail height), tapering gradually to a blunt end. Color in formaldehyde solution is dark brown with greyish tinge; subtle light stippling on the sides and dorsally uniform. Upper lip margin shows a narrow frontal line of whitish pigmentation which does not reach the nostrils. A lighter pale line runs distal and sagittally all along the dorsal surface of the tail. Underside is pale beige, brighter and spotless under head, cloaca, tail, and undersides of hands and legs. The flanks of the belly and the underside of the arms show obscure brown blotches. A big dark blotch covers the underside of the left hindlimb around the ankle. The lower half of the abdomen is open as a result of a necropsy. Dorsal and ventral pictures in Fig. 5, and morphometric measurements shown in Table 7.

Figure 5 Designated holotype and paratypes of the Western Montseny brook newt Calotriton arnoldi laietanus ssp. nov.

The specimens MZB 2024-2573, MZB 2024-2574 and MZB 2024-2575 are deposited at MCNB. The specimen MCNG 13809 is deposited at MCNG.

Table 7 Description measurements of Calotriton arnoldi laietanus spp. nov. holotype and paratypes.

Measurements	MZB 2024-2573	MZB 2024-2574	MZB 2024-2575	MCNG13809	
SVL	60.26	59.62	57.59	48.07	
HL	13.49	13.67	15.32	11.68	
FLL	13.20	12.34	12.37	10.81	
HLL	12.94	13.32	10.96	10.00	
LI	26.94	26.23	27.83	23.93	
TL	45.65	44.35	37.64	34.64	
TH	7.51	5.79	8.46	4.66	
DNOS	3.43	3.23	3.58	2.84	
DORB	4.29	5.15	5.17	4.05	
DES	2.86	3.25	3.04	2.67	
DEP	8.47	7.49	10.34	6.84	
TW	4.12	3.15	4.17	3.04	
Note:

MZB 2024-2573 (holotype), MZB 2024-2574 and MZB 2024-2575 are deposited at Museu de Ciències Naturals de Barcelona, and MCNG 13809 at Museu de Ciències Naturals de Granollers. SVL, snout-vent length; HL, head length; FLL, forelimb length; HLL, hindlimb length; LI, limb interval; TL, tail length; TL, tail height; DNOS, distance between nostrils; DORB, minimum distance between inner angle of eye orbits; DES, distance from anterior angle of the eye orbit to the snout; DEP, distance from the posterior angle of the eye orbit to the posterior margin of parotid area; TW, maximum tail width.

Variation

All living and preserved adult specimens of the Western Montseny brook newt lack yellow blotches from the tail and body flanks. Underside pale coloration of the body varies among the type specimens, being almost as dark as the flanks of the body in MZB 2024-2574. Tails vary from almost not dorsally expanded and homogeneous along all their length, like in the female MZB 2024-2574, to wider and deeper tails, such as in the male MZB 2024-2575. Tail tips have been clipped in specimens MZB 2024-2575 and MZB 2024-2574. MZB 2024-2575 exhibits an obvious white upper lip, with white pigmentation reaching the nostrils. Skin of anterior parts of the body and head had been ripped off in individual MCNG 13809, which was found dead in the wild after a storm. A necropsy was performed on specimen MZB 2024-2575, which has an abdominal incision all along its body as a result. Dorsal and ventral pictures of the paratypes are shown in Fig. 5, and morphometric measurements in Table 7.

Discussion

Ecological and morphological divergence

We present the first evidence of ecological differences between the ranges of Montseny brook newt lineages, here formally described as distinct subspecies. These differences largely relate to contrasting forest assemblages (whose resolution was, in addition, more suitable for the scale of this study), and are hypothesized to explain previously described color variation between lineages (Valbuena-Ureña, Amat & Carranza, 2013). Eastern Montseny brook newts often exhibit light yellowish blotches on the sides of the tail and body, perhaps providing a better camouflage against the background formed by the abundant brown or yellowish leaf litter that beds their brooks, covered by deciduous forests (Fig. 3). On the other hand, the Western Montseny brook newts occupy brooks with scarce leaf litter in evergreen forests, exhibiting more cryptic coloration (darker, without yellow blotches), and often displaying whitish snout margins, that may resemble the underside of holm oak leaves (Figs. 2A and 3). If these color differences are adaptive, the yellowish blotched pattern might be the ancestral form for the species, as some Western juveniles faintly and temporarily develop it as well.

Additionally, our morphological analyses considering shape independently from size yielded more diagnostic patterns, such as significant differences in tail height between lineages, instead of sexual dimorphism-related, as previously reported (Valbuena-Ureña, Amat & Carranza, 2013). Despite the broad overlap among linear measurements, morphological PC2 (accounting for 16.49% of the shape variance) clearly depicted shape differences between Western and Eastern Montseny brook newts, regardless of sex (Fig. 2C). Overall, Western Montseny brook newts are more robust, with shorter limbs and wider heads (Fig. 2A), perhaps because of a higher dependence on hiding in rock interstices instead of among leaf litter. On the other hand, Eastern Montseny brook newts have longer and more dorso-ventrally expanded tails, which may be advantageous when facing the stronger currents in the brooks they inhabit, characterized by higher altitudes and precipitation and steeper slopes.

If these color and shape differences are a response to diverging ecological pressures, it is intriguing how selection has proceeded in such a short timeframe (divergence among lineages dates back roughly to the Last Interglacial, 110 kya, Talavera et al. (2024)). In small-sized populations, genetic drift becomes a major evolutionary force, and its importance compared to selection increases (Whitlock, 2000). Thus, beneficial alleles are more likely to be lost and random alleles more likely to reach high frequencies, changing faster as effective population sizes decrease (Whitlock, 2004). This scenario may well apply to the Montseny brook newt, whose populations have declined, at least, since the beginning of the Late Glacial Interstadial (c. 14,670 kya, Talavera et al., 2024). From then on, a series of bottlenecks drastically reduced the effective population size of the species, while very low levels of historical migration between submassifs promoted fixation and genomic divergence (Talavera et al., 2024). In this high-drift scenario, the role of selection seems virtually negligible, and would imply that small populations are doomed to extinction due to the accumulation of deleterious alleles; however, sexual selection can act to reduce the fixation probability of deleterious mutations and increase the probability of fixation of new beneficial mutations (Whitlock, 2000). Negative assortative mating is a form of sexual selection in which individuals prefer to mate with less similar (in phenotype or genotype) counterparts, and has been invoked to explain higher values of observed versus expected heterozygosity in natural populations of the Montseny brook newt (Valbuena-Ureña et al., 2017a; Talavera et al., 2024). Under this mating strategy, the effective population size virtually increases, offering then higher probabilities for beneficial alleles to be selected and to avoid the lethal accumulation of deleterious mutations.

Taxonomy and its implications on conservation

In light of the morphological, genomic, and ecological evidence of the distinctness of the two Montseny brook newt lineages, we deemed these lineages to deserve recognition as different subspecies. Intraspecific lineages identified in phylogeographic studies but lacking a formal taxonomic rank—such as the Western Montseny lineage of C. arnoldi prior to this study—are rarely considered in conservation policies, taxonomic lists and biodiversity accounts (Hoban et al., 2021). Nevertheless, species conceptualization and delimitation are controversial topics amongst biologists, and so is the subspecies definition (e.g., Braby, Eastwood & Murray, 2012; Burbrink et al., 2022; De Queiroz, 2021, 2020; Hillis, 2020, 2021; Padial & De la Riva, 2021). Under the updated subspecies concept of De Queiroz (2020), subspecies are incompletely separated “species”, supported by the same kinds of evidence that would be required to infer that an entity is a species, as well as evidence that its separation from other species is incomplete. In this case, several secondary criteria for supporting a subspecies or species are met: genetic and genomic reciprocal monophyly, diagnosability (color) and differences in ecology and morphology. Furthermore, evidence of past inter-lineage migration has been reported (Talavera et al., 2024) and, therefore, reproductive isolation is unlikely despite their currently allopatric ranges. As subspecies according to De Queiroz (2020) represent another type of species, trinomials “can (but not need to)” be used to indicate the nesting of incompletely separated lineages within a more inclusive lineage. Although we agree that this point of view better depicts the evolutionary continuum of speciation, here we stick to the subspecies definition by Hillis (2020, 2021), using trinomials categorically to indicate incompletely separated lineages within species. This definition agrees with Dufresnes, Poyarkov & Jablonski (2023) as well, who, furthermore, propose using the divergence time as a proxy of reproductive isolation between lineages in which reproductive barriers to geneflow are untestable due to the lack of secondary contacts. This is the case of the Montseny brook newt lineages, which, in addition, have never been kept together in captivity. If reproductive isolation evolves through the gradual accumulation of molecular incompatibilities between diverging lineages (Dufresnes et al., 2021), we can hypothesize that their divergence is too recent to have achieved strong postzygotic isolation mechanisms.

Beyond the different species/subspecies concepts and definitions, an often-overlooked point regards the implications of taxonomic changes on conservation policies. As pointed out by Garnett & Christidis (2017), conservation legislation frequently fails to keep pace with changes in how species are named. Thus, finer taxonomic splitting could make certain taxa more vulnerable, for example, leaving newly described species exposed to illegal trade until their formal inclusion in conservation policies (e.g., Zhou et al., 2016). The same type of ambiguity arises if there is no consensus with regard to the use of binomials or trinomials for subspecies, as proposed by De Queiroz (2020). On the other hand, as May (1990) warned, “bad taxonomy can kill”, i.e., some cryptic taxa may go extinct due to a lack of awareness of their existence and conservation requirements. This omen was confirmed throughout the years, with grievous examples, such as the Chinese giant salamanders (Yan et al., 2018) (but see Morrison et al. (2009)). As the absence of taxonomic designation hampers conservation and unnamed lineages face higher extinction risk than described taxa (Liu et al., 2022; Dufresnes, Poyarkov & Jablonski, 2023), we recognize the uniqueness of the Western Montseny brook newt by providing it with a formal name. Finally, as advocates of the use of trinomial names, we must point out that the use of the subspecific rank is currently increasing among European amphibians, with many examples of their application based on genetic or genomic evidence (e.g., in the genera Salamandra (Šunje et al., 2019; Burgon et al., 2021), Triturus (Arntzen, 2023), Ichthyosaura (Vörös et al., 2021), Chioglossa (Sequeira et al., 2022), Alytes (Dufresnes & Hernandez, 2021) and Rana (Dufresnes et al., 2023)). This re-popularization of the subspecies is in line with a deeper understanding of complex evolutionary histories, such as those shaped by the Pleistocene glacial cycles in Europe, only fully unveiled with the advent of genomics. Genomic taxonomy, therefore, allows researchers to implement more integrative, useful, and stable taxonomic rearrangements (Dufresnes, Poyarkov & Jablonski, 2023).

Conclusions

The Montseny brook newt (Calotriton arnoldi) inhabits a small but fragmented range in northeast Spain. Genetic and genomic differences supporting the existence of two lineages is complemented in this study with ecological and phenotypical evidence of diverging evolutionary trajectories in this Critically Endangered species. Taxonomy plays a key role in conservation, as unnamed lineages are more prone to extinction than formally described taxa (Liu et al., 2022). However, in spite of the current rapid advances in methods and the towering production of genomic data, evolutionary biologists are often reluctant to explicitly propose taxonomic rearrangements. Paradoxically, we are witnessing a phenomenon of species inflation, as a result of a general shift towards a “one level” taxonomy based only on species and neglecting the subspecies rank (Dufresnes, Poyarkov & Jablonski, 2023). Taxonomic instability can lead to legislation ambiguities and gaps, resulting in valid taxa lacking legal protection. In this context, we aim to raise awareness of conservation needs of the Western Montseny brook newt C. a. laietanus ssp. nov. as a separate entity, formalizing their de facto treatment as a different management unit by the conservation program, and without exposing the new taxon to a legal vacuum due to the ambiguities of binomial taxonomy.

Supplemental Information

Supplemental Information 1 Ecology dataset.

Environmental, altitude and vegetation variables from 596 location in which C. arnoldi has been found. Coordinates are not disclosed for conservation reasons.

Supplemental Information 2 Morphology dataset.

Filtered dataset used in this study, originally published by Valbuena-Ureña et al. (2013).

We must thank three anonymous reviewers for their constructive comments provided on a previous version of the manuscript, as well as Loukia Spilani, Maria Estarellas, Juliana Tabares, Laia Pérez-Sorribes, Elena Obon, Mónica Alonso, Sonia Solorzano, Daniel Guinart, Albert Montori and Daniel Fernández-Guiberteau for their help and support. The study of the distribution of C. arnoldi was developed thanks to the collaboration of the Agents Rurals from Generalitat de Catalunya.

Additional Information and Declarations

Competing Interests

Author Contributions

Data Availability

New Species Registration

The authors declare that they have no competing interests.

Adrián Talavera conceived and designed the experiments, performed the experiments, analyzed the data, prepared figures and/or tables, authored or reviewed drafts of the article, and approved the final draft.

Emilio Valbuena-Ureña conceived and designed the experiments, performed the experiments, analyzed the data, authored or reviewed drafts of the article, and approved the final draft.

Bernat Burriel-Carranza performed the experiments, authored or reviewed drafts of the article, and approved the final draft.

Gabriel Mochales-Riaño performed the experiments, authored or reviewed drafts of the article, and approved the final draft.

Dean C. Adams conceived and designed the experiments, analyzed the data, authored or reviewed drafts of the article, and approved the final draft.

Fèlix Amat conceived and designed the experiments, performed the experiments, authored or reviewed drafts of the article, and approved the final draft.

Francesc Carbonell performed the experiments, authored or reviewed drafts of the article, and approved the final draft.

Salvador Carranza conceived and designed the experiments, performed the experiments, analyzed the data, authored or reviewed drafts of the article, and approved the final draft.

The following information was supplied regarding data availability:

The raw data for ecological and morphological analyses are available in the Supplemental Files. The morphology dataset is from Valbuena-Ureña, Amat & Carranza (2013), although juvenile individuals have been removed.

The following information was supplied regarding the registration of a newly described species:

Publication LSID: urn:lsid:zoobank.org:pub:AE201B51-F9B7-4B7F-A70E-D5FCC05EB335

Calotriton arnoldi laietanus ssp. nov. LSID: urn:lsid:zoobank.org:act:5C8988C2-68EE-4AC6-8F67-8602FAA84CD2.

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
