# Peer review of "Integrative systematic revision of the Montseny brook newt (Calotriton arnoldi), with the description of a new subspecies"

_PeerJ, doi:10.7717/peerj.17550_

## Round 0.1 · original submission · Major Revisions

Thank you for submitting your paper to PeerJ. This paper has been reviewed by three experts in the field and all three have said that your paper is clear, well-written and well-described. One reviewer pointed out that there is an important issue that needs to be addressed before this paper can be published, which is the holotype specimen chosen. The specimen chosen is not in the best shape and because this species is in captive colonies, a fresher specimen could be used instead, or in conjunction with the specimen used here.

I look forward to reading your edited manuscript!

Reviewer 1 ·

Basic reporting

Talavera and colleagues describe a new subspecies of the Critically Endangered Montseny brook newt based on genetic, ecological and morphological differentiation. This is a straightforward paper that is easy to follow.

Experimental design

There is one point I would like to raise that I think the authors should acknowledge: given the tiny range, the spatial resolution of the bioclim values seems huge in comparison. By the way, it is good to see that no animal had to be sacrificed for the description.

Validity of the findings

Taxonomy is of course very much a matter of opinion, but the author's conclusion is based on a variety of data that supports their choice. I also enjoy the discussion on the subspecies concept.

Additional comments

Minor language issues:
Line 75: fish predators are predators of fish, not fish that are predators
Line 338: unclear what This refers to, I guess size reduction of population size but could be interpreted as beneficial alleles being lost (for which there is no evidence).

·

Basic reporting

no comment

Experimental design

no comment

Validity of the findings

no comment

Additional comments

This article constitutes a great scientific work and is of primary interest in conservation biology. It perfectly completes the work already published (Talavera et al., 2024) and offers an excellent argument for considering subspecies in Calotriton arnoldi.

Reviewer 3 ·

Basic reporting

The paper describes a new subspecies of a Critically Endangered newt species based on new ecological data and the reanalysis of published morphometric data. The new taxon is also supported by genomic evidence. The manuscript is well-written, figures and tables are sufficient and well presented, and previous studies have been adequately referenced. I have made some edits in the manuscript suggesting small changes in the text for clarity that I hope will be useful to the authors. I can provide a Word file with the changes suggested for convenience (the website only allows uploading a PDF).

Experimental design

The new data and analyses are sound and adequately illustrated and discussed.

Validity of the findings

The new findings are relevant and the conclusions, including the description of a new taxon, are justified and supported by the data.

Additional comments

My only serious concern is the choice of the holotype, which is a very poorly preserved specimen. This is not acceptable and I urge the authors to choose another specimen, ideally in excellent preservation condition. To me this is a critical point that needs to be addressed before the paper can be accepted for publication

Annotated reviews are not available for download in order to protect the identity of reviewers who chose to remain anonymous.

---

## Round 0.2 · Minor Revisions

Thank you for submitting this updated version of the manuscript. One reviewer had minor comments that should be addressed before publication. I look forward to your revised manuscript

Reviewer 1 ·

Basic reporting

no comment

Experimental design

no comment

Validity of the findings

no comment

Additional comments

The authors have revised their manuscript to my satisfaction

Reviewer 3 ·

Basic reporting

My main concern with the previous version was the choice of a specimen as the holotype, which has been satisfactorily fixed. Other than that, I provided suggestions to improve the text that have been largely ignored, with no explanation. For instance, in the Abstract, I don't understand the use of "Despite" on L27, since a range can be divided by a deep valley regardless of whether the distribution is restricted or not; also I don't think populations can "behave as lineages". There are many other changes, like maintaining "mountains ranges" (L45), which is incorrect, but I don't have the time to go again through all this, so I leave it to the editor and authors to edit the text as they wish.

Experimental design

No comment

Validity of the findings

No comment

Additional comments

No comment

---

## Round 0.3 · accepted · Accept

Thank you for your most recent revision. You have satisfied all reviewer comments and this manuscript is now ready for publication. Congratulations!